# Loss of *Mfn1* but not *Mfn2* enhances adipogenesis

Jake P. Mann[1], Luis Carlos Tábara[2], Satish Patel[1], Pushpa Pushpa[1], Anna Alvarez-Guaita[1], Liang Dong[1], Afreen Haider[1], Koini Lim[1], Panna Tandon[3], Fabio Scurria[1], James E. N. Minchin[3], Stephen O'Rahilly S.[1], Daniel J. Fazakerley[1], Julien Prudent[2], Robert K. Semple[3,4], David B. Savage[1]*

1 Metabolic Research Laboratories, Wellcome Trust-Medical Research Council Institute of Metabolic Science, University of Cambridge, Cambridge, United Kingdom, 2 MRC Mitochondrial Biology Unit, University of Cambridge, Cambridge, United Kingdom, 3 Centre for Cardiovascular Science, University of Edinburgh, Edinburgh, United Kingdom, 4 MRC Human Genetics Unit, University of Edinburgh, Edinburgh, United Kingdom

* dbs23@cam.ac.uk

**Data Availability Statement:** RNA sequencing data is available: https://www.ncbi.nlm.nih.gov/geo/query/acc.cgi?acc=GSE232502.

**Funding:** J.P.M. is supported by a Wellcome Trust fellowship (WT 216329). R.K.S, D.B.S. and S.O'R.

## Abstract

### Objective

A biallelic missense mutation in mitofusin 2 (*MFN2*) causes multiple symmetric lipomatosis and partial lipodystrophy, implicating disruption of mitochondrial fusion or interaction with other organelles in adipocyte differentiation, growth and/or survival. In this study, we aimed to document the impact of loss of mitofusin 1 (*Mfn1*) or 2 (*Mfn2*) on adipogenesis in cultured cells.

### Methods

We characterised adipocyte differentiation of wildtype (WT), *Mfn1*[−/−] and *Mfn2*[−/−] mouse embryonic fibroblasts (MEFs) and 3T3-L1 preadipocytes in which Mfn1 or 2 levels were reduced using siRNA.

### Results

*Mfn1*[−/−] MEFs displayed striking fragmentation of the mitochondrial network, with surprisingly enhanced propensity to differentiate into adipocytes, as assessed by lipid accumulation, expression of adipocyte markers (*Plin1*, *Fabp4*, *Glut4*, *Adipoq*), and insulin-stimulated glucose uptake. RNA sequencing revealed a corresponding pro-adipogenic transcriptional profile including *Pparg* upregulation. *Mfn2*[−/−] MEFs also had a disrupted mitochondrial morphology, but in contrast to *Mfn1*[−/−] MEFs they showed reduced expression of adipocyte markers. *Mfn1* and *Mfn2* siRNA mediated knockdown studies in 3T3-L1 adipocytes generally replicated these findings.

### Conclusions

Loss of *Mfn1* but not *Mfn2* in cultured pre-adipocyte models is pro-adipogenic. This suggests distinct, non-redundant roles for the two mitofusin orthologues in adipocyte differentiation.

are supported by the Wellcome Trust (WT 210752, WT 219417 and WT 214274 respectively), the MRC Metabolic Disease Unit, the National Institute for Health Research (NIHR) Cambridge Biomedical Research Centre and the NIHR Rare Disease Translational Research Collaboration (https://www. mrl.ims.cam.ac.uk/mrc-metabolic-diseases-unit/) MRC_MC_UU_12012.1. J.P. is supported by the Medical Research Council (MRC) (MRC grants MC_UU_00015/7 and MC_UU_00028/5) and a Biotechnology and Biological Sciences Research Council (BBSRC) research grant (BB/W008467/1). L.C.T. is supported by a BBSRC research grant (BB/W008467/1). J.E.N.M. is supported by a MRC New Investigator Grant (MR/S025685/1) and D.J.F. is supported by a Medical Research Council Career Development Award (MR/S007091/1). The funders had no role in study design, data collection and analysis, decision to publish, or preparation of the manuscript. Funding to the Genomics and Transcriptomics Core is from the MRC Metabolic Diseases Unit [MC_UU_00014/5] and the Wellcome Trust Major Award [208363/Z/17/Z]. The funders had no role in study design, data collection and interpretation, or submitting the work for publication.

**Competing interests:** The authors have declared that no competing interests exist.

## Introduction

Primary mitochondrial disorders result in pleiotropic syndromes disturbing the function of multiple organ systems commonly including the central and peripheral nervous system and neuromuscular function [1, 2]. Adipose tissue dysfunction and related features of the metabolic syndrome are not present in most primary mitochondrial disorders [3, 4]. One striking exception to this is the recent discovery that a point mutation in human mitofusin 2 (*MFN2*) produces profound adipose tissue dysfunction, non-alcoholic fatty liver disease, insulin resistance and type 2 diabetes [5–7]. Humans with biallelic p.Arg707Trp MFN2 mutations develop increased upper body adiposity (multiple symmetric lipomatosis) with lower body lipodystrophy. Over-grown adipose depots show deranged ultrastructure of the mitochondrial network but largely normal adipocytes on light microscopy [5]. Affected adipose tissue exhibits a strong transcriptomic signature of activation of the integrated stress response and increased pro-growth signalling. This adipose phenotype appears to be specific to the p.Arg707Trp mutation, which has been identified in all reported cases to date. Other *MFN2* mutations, many effectively leading to loss of one allele (i.e. haploinsufficiency) and often located in the N-terminal GTPase domain, are associated with autosomal dominant sensorimotor neuropathy without an adipose phenotype [8]. No monogenic disease attributable to *MFN1* mutations has yet been reported.

Within the last 20 years, the dynamic nature of the mitochondrial network has been clearly established, and many of the molecules mediating mitochondrial morphology have been identified [9, 10]. Mitochondrial network dynamics are entrained to adapt to the cellular nutritional state: starvation or nutrient excess trigger changes in mitochondrial morphology that correspond to functional changes in mitochondrial oxidative phosphorylation [11] and the generation of reactive oxygen species (ROS) [12, 13]. The mitofusins (*MFN1* and *MFN2*) are GTPases required for fusion of the outer mitochondrial membrane. Loss of either mitofusin leads to fragmentation of the mitochondrial network and disrupts its function [14–16]. *Mfn1* is widely expressed and is required for mitochondrial fusion whereas *Mfn2* reportedly has additional roles, such as being intimately involved in forming membrane contact sites between mitochondria and the endoplasmic reticulum [17] or lipid droplets [18], and in facilitating apoptosis and mitophagy [19].

Global *Mfn1* or *Mfn2* deletion in mice is embryonically lethal, but knockout of *Mfn2* [20–22] specifically in mature adipocytes has illustrated the importance of mitochondrial network dynamics in adipose tissue. Loss of *Mfn2* from *Ucp1*-expressing brown adipose tissue impairs its thermogenic capacity and increases lipid droplet size [20, 21], but surprisingly increases insulin sensitivity. In contrast, deletion of *Mfn2* from mature white adipocytes in adult mice was obesogenic and impaired insulin sensitivity [22]. Adipose-specific knock-out of *Mfn1* has yet to be described. Importantly, all promoters used for adipose-specific knockouts to date have been from genes expressed during terminal adipocyte differentiation, so whether loss of one or both mitofusins perturbs the process of adipogenesis, which involves significant expansion of mitochondrial mass [23–25], is unknown.

Genetic perturbation of mitochondrial function has been shown to cause loss of white adipose tissue (lipoatrophy) in some in vivo models, providing further evidence for the importance of mitochondrial function in adipose function. Adipose-tissue specific (using adiponectin-Cre) loss of *Tfam*, a key regulator of mitochondrial biogenesis, caused a classical lipodystrophic phenotype: loss of adipose mass and insulin resistance [26]. However, other models have resulted in lipoatrophy without insulin resistance [27] or even increased glucose tolerance [28, 29].

Impaired cold adaptation is a fairly consistent feature of models of mitochondrial dysfunction in adipose tissue, particularly where mitochondrial function is perturbed in brown

adipose tissue alone or in addition to perturbation in white adipose tissue. Loss of *Mfn2* in all adipose tissue (adiponectin-Cre) [20] or brown adipose tissue (Ucp1-Cre) [21] both cause cold intolerance. Similarly, adipose tissue-specific loss of *Opa1* [30], which mediates fusion of the inner mitochondrial membrane, causes cold intolerance. This underscores the prominent role of mitochondrial dynamics in cold-induced thermogenesis.

Differentiation of fibroblast-like cells into mature adipocytes is a well-established experimental paradigm used to dissect molecular regulation of adipogenesis [31, 32]. In this study we examined the role of *Mfn1* and *Mfn2* in adipogenesis using knock-out mouse embryonic fibroblasts (MEFs), validating findings in 3T3-L1 preadipocytes using siRNA-mediated knock-down. We found that loss of *Mfn1* enhances adipogenesis, whereas loss of *Mfn2* has no effect on total lipid accumulation but alters lipid droplet morphology and reduces the expression of some adipogenic markers. These findings suggest that the two mitofusin proteins have important but divergent roles in the control of adipogenesis.

## Methods

### Culture and adipogenic differentiation of MEFs

Mouse embryonic fibroblasts (MEFs) null for $Mfn1^{-/-}$, $Mfn2^{-/-}$, $Mfn1^{-/-}2^{-/-}$, and $Opa1^{-/-}$, and wild-type MEFs, originally derived by Chen *et al.* [14], were obtained for use in this study from American Type Culture Collection in February 2019 (see S1 Table for a list of reagents).

MEFs were cultured in Dulbecco's Modified Eagle's Medium supplemented with 10% (vol/vol) foetal bovine serum (FBS), 1% (1mM) sodium pyruvate (from stock 100mM), 1% non-essential amino acids 100x, 1% penicillin/streptomycin (10,000 units penicillin and 10 mg streptomycin per mL), 2mM L-glutamine, and 50mM 2-mercaptoethanol. All cells were grown at 37°C in 20% $O_2$ and 5% $CO_2$.

Cells were maintained and passaged whilst sub-confluent prior to adipogenic differentiation. All experiments were performed with cells at passage <10. To initiate differentiation, cells were grown to confluence (day -2) in 6- or 12-well plates and then, two days later, treated with 8μg/mL D-pantothenic acid, 8μg/mL biotin, 1μM rosiglitazone, 0.5mM 3-isobutyl-1-methylxanthine (IBMX), 1μM dexamethasone, and 1μM insulin. Treatment with D-pantothenic acid, biotin, rosiglitazone, and insulin was repeated on day +2, +4, and +6. Differentiation continued until day +8.

For fluorometric quantification of intracellular lipid accumulation, cells were incubated with AdipoRed at 1:25 dilution and 37°C for 20 minutes in the dark. Cells were washed twice in PBS then fluorescence was measured on Tecan M1000 Pro plate reader with excitation at 485nm and emission at 572nm.

For studies using the reactive oxygen species (ROS) scavenger N-acetylcysteine (NAC), MEFs were differentiated according to the above protocol in the presence of pH7 adjusted NAC (Sigma) at either 1mM, 2.5mM, or 5mM. NAC treatment began at day -2 and was repeated with each media change.

### Over-expression of *Pparg2* in MEFs

To improve the adipogenic potential of MEFs, cells were retrovirally transduced with murine Peroxisome proliferator-activated receptor gamma 2 (*Pparg2*). Phoenix-AMPHO (ATCC) cells were used for packaging. 70% confluent Phoenix cells were transfected with 12μg pBABE (puromycin-resistant)-mPparg2 vector or pBABE-EGFP (control) plasmid DNA using Lipofectamine 3000 transfection reagent. 72-hours later the supernatant media containing the virus was collected and filtered through a 0.45μm membrane. Retroviral stocks were used to transduce 50–60% confluent MEFs with addition of 12μg/mL polybrene. From 24-hours after

transduction, puromycin selection began using 4μg/mL, and MEFs were subsequently cultured in media containing this concentration of puromycin. Adipogenic differentiation was then performed according to the above protocol.

## Confocal microscopy of mitochondrial network

Cells were stained with Mitotracker Orange CMTMRos (ThermoFisher) for imaging of the mitochondrial network using confocal microscopy. MEFs were grown on cover slips to 80–90% confluence (pre-adipocytes at day -2 for MEFs and day 0 for 3T3-L1s) and treated with 400nM Mitotracker for 20 minutes at 37°C in the dark. Cells were washed with fresh media four times, each for 30 minutes at 37°C in the dark, followed by three washes in phosphate buffered saline (PBS). Cells were then fixed in 4% formaldehyde at room temperature for 10 minutes and mounted onto slides using ProLong Gold or Diamond Antifade Mountant with 4′,6-diamidino-2-phenylindole (DAPI) (ThermoFisher). Images were obtained using a Leica SP8 confocal microscope using excitation/emission spectra: Mitotracker Orange CMTMRos (550/560-580), BODIPY (490/515-535), and DAPI (405/450-470).

## Culture and adipogenic differentiation of 3T3-L1 fibroblasts

Mouse fibroblast 3T3-L1 fibroblasts (3T3-L1s) [33] were used as an in vitro model of adipocyte differentiation and were obtained as a gift from D.J. Fazakerley and D.E. James (University of Sydney), originally obtained from Dr. Howard Green (Harvard Medical School, Boston, MA). 3T3-L1s were cultured in Dulbecco's Modified Eagle's Medium supplemented with 10% (vol/vol) foetal bovine serum and 1% GlutaMax. Cells were grown at 37°C in 20% $O_2$ and 10% $CO_2$. Cells were maintained and passaged sub-confluent prior to adipogenic differentiation.

To initiate differentiation, cells were grown to confluence (day -2) in 12-well plates and then two days later (day 0) treated with 0.5mM IBMX, 410nM biotin, 220nM dexamethasone, and 350nM insulin. After 72-hours (day +3), media was replaced with addition of 350nM insulin. A further 72-hours later (day +6) media was changed (without any additional chemicals) and subsequently replaced on day +8 and day +10 with experiments ending on day +12. On day +12, lipid accumulation was assessed using Oil Red O staining and AdipoRed quantification. RNA and protein were also extracted at day +12.

## siRNA knock-down experiments in 3T3-L1s

For knock-down, confluent pre-adipocyte (day -2) 3T3-L1s grown in 12-well plates were transfected with a pool of two anti-*Mfn1* siRNAs (ThermoFisher Scientific, CatIDs: s85002 & s85004), each at 40nM to achieve a final concentration of 80nM, with 2.5μl RNAiMax (ThermoFisher Scientific) in OptiMEM (ThermoFisher Scientific). In addition, we tested the knock-down efficacy of a third anti-*Mfn1* siRNA (ThermoFisher Scientific, CatID s85003), which had limited efficacy and was not taken forward in further experiments. For knock-down of *Mfn2*, an anti-*Mfn2* SmartPool (Dharmacon) was used at a final concentration of 40nM. In all experiments, a scrambled Silencer Negative Control siRNA (ThermoFisher Scientific) was used at 80nM. Knock-down was performed with each media change, i.e. day -2, 0, +3, +6, +8, and +10.

## Immunofluorescence in 3T3-L1s

Mature (day +8) differentiated 3T3-L1 adipocytes were reseeded onto matrigel-coated (Corning) coverslips at 50% density. siRNA knockdown was performed during reseeding. Reseeded adipocytes (day 9) were cultured under standard "high glucose" conditions (Dulbecco's

modified eagle's medium containing 25mM glucose supplemented with 10% (vol/vol) foetal bovine serum and 1% GlutaMax) or "low glucose" (serum-free Dulbecco's modified eagle's medium containing 5mM glucose supplemented with 0.2% (w/v) bovine serum albumin and 1% GlutaMax) conditions for 16 hours. Cells were then fixed in 4% paraformaldehyde (PFA) in PBS at room temperature for 20 minutes, quenched with 100mM glycine in PBS, washed three times with PBS, followed by incubation with 50mM ammonium chloride in PBS to quench the unspecific fluorescence signal from aldehyde groups. Cells were washed again three times with PBS and permeabilized in 0.1% Triton X-100 in PBS for 10 minutes. Then, cells were blocked with 10% FBS in PBS, followed by incubation with the appropriate primary antibodies in 5% FBS in PBS, for two hours at room temperature. After three washes with 5% FBS in PBS, cells were incubated with specific secondary antibodies (1:1,000) for one hour at RT. After three washes in PBS, coverslips were mounted onto slides using Dako fluorescence mounting medium (Dako).

Mitochondria were stained using a rabbit anti-TOMM20 antibody (11802-1-AP/Protein-tech) (1:1,000). Alexa Fluor 488 (anti-rabbit) was used as a secondary antibody (1:1,000) (Invitrogen). Lipid droplets were stained by incubating coverslips with LipidTOX deep red (Thermo-fisher) for 2 hours (1:1,000).

For confocal image acquisition, cells were visualized using a 100X objective lenses (NA1.4) on a Nikon Eclipse TiE inverted microscope and 7 stacks of 0.2μm each were acquired with appropriate lasers using an Andor Dragonfly 500 spinning disk system, equipped with a Zyla 4.2 PLUS sCMOS camera (Andor), coupled with Fusion software (Andor). Images from the same experiment were acquired following the same parameters including exposure time and laser intensities.

For image analysis, mitochondrial morphology was manually analysed and classified as intermediate, hyperfused or fragmented as already described [34]. The different mitochondrial parameters (area, length and number) were quantified by randomly selecting region of interests (ROIs) of 225μm$^2$ at the cell periphery and analysed using MitoMapr (Fiji) [35]. Error bars displayed on graphs represent the mean ± S.D from 3 independent experiments (at least 18 cells per experiment and condition). Statistical significance was analysed using Nested t-test or Two-way ANOVA test using GraphPad Prism software. *$p < 0.05$, **$p < 0.01$, ***$p < 0.001$ and ****$p < 0.0001$ were considered significant.

## Electron microscopic imaging of mitochondria

Transmission electron microscope (TEM) studies were performed at the Cambridge Advanced Imaging Centre, University of Cambridge. Undifferentiated MEFs (day -2) were grown to sub-confluence in 6-well plates. Cells were washed twice in 0.9% NaCl then fixed in (2% glutaraldehyde/2% formaldehyde in 0.05M sodium cacodylate buffer pH 7.4 containing 2mM calcium chloride) for 4 hours at 4˚C. Cells were then scraped from plates and, after washing five times with 0.05M sodium cacodylate buffer pH 7.4, samples were osmicated (1% osmium tetroxide, 1.5% potassium ferricyanide, 0.05M sodium cacodylate buffer pH 7.4) for three days at 4˚C. After washing five times in DIW (deionised water), samples were treated with 0.1% (w/v) thio-carbohydrazide/DIW for 20 minutes at room temperature in the dark. After washing five times in DIW, samples were osmicated a second time for 1 hour at room temperature (2% osmium tetroxide/DIW). After washing five times in DIW, samples were block stained with uranyl acetate (2% uranyl acetate in 0.05M maleate buffer pH 5.5) for three days at 4˚C. Samples were washed five times in DIW and then dehydrated in a graded series of ethanol (50%/70%/95%/100%/100% dry) 100% dry acetone and 100% dry acetonitrile, three times in each for at least 5 minutes. Samples were infiltrated with a 50/50 mixture of 100% dry acetonitrile/

Quetol resin (without benzyldimethylamine (BDMA)) overnight, followed by three days in 100% Quetol (without BDMA). Then, samples were infiltrated for 5 days in 100% Quetol resin with BDMA, exchanging the resin each day. The Quetol resin mixture is: 12g Quetol 651, 15.7g NSA, 5.7g MNA and 0.5g BDMA (all from TAAB). Samples were placed in embedding moulds and cured at 60˚C for 3 days.

Thin sections were cut using an ultramicrotome (Leica Ultracut E) and placed on bare 300 mesh copper TEM grids. Samples were imaged using a Tecnai G2 TEM (FEI/Thermo Fisher Scientific) run at 200 keV using a 20μm objective aperture to improve contrast. Images were acquired using an ORCA HR high resolution CCD camera (Advanced Microscopy Techniques Corp, Danvers USA).

Analysis was performed by manual measurement of individual mitochondria from all obtained images using ImageJ. Despite multiple mitochondria measured per sample, statistical tests (pairwise comparisons relative to wild-type) were based on the mean from two independent biological replicates and false-discovery rate adjustment for the number of tests [36].

## mtDNA content assay

Relative mitochondrial DNA (mtDNA) content was assayed using real-time quantitative polymerase chain reaction (RT-qPCR) for a mitochondrial DNA gene (*Rnr2*) and a nuclear DNA gene (*Hk2*). DNA was extracted from undifferentiated MEFs at 80% confluence using DNeasy kit (Qiagen). DNA was quantified on a Nanodrop and diluted to 4ng/μl. RT-qPCR was performed in triplicate for each sample using 8ng DNA with primers for *Hk2* (Forward: GCCAGCCTCTCCTGATTTTAGTGT, Reverse: GGGAACACAAAAGACCTCTTCTGG) and *Rnr2* (Forward: AACTCGGCAAACAAGAACCC, Reverse: CCCTCGTTTAGCCGTTCATG). The ratio mt*Rnr2*/n*Hk2* was calculated using the standard curve method and expressed relative to wild-type.

## Oil Red O staining

Mature adipocytes (day +8 MEFS or day +12 3T3-L1s) in six-well plates were gently washed in PBS before fixing in 10% formalin for 30 minutes at room temperature. Cells were then washed twice in PBS before dehydration in 60% isopropanol for 5 minutes (twice). Oil Red O (ORO) working solution was then added for 20 minutes. ORO working solution was made from 1g ORO (Sigma) dissolved in 400ml isopropanol, diluted 3:2 with milli-Q water and filtered through a 0.45μm membrane. Cells were washed with water then imaged on a flat-bed scanner and with light microscopy. Lipid droplet width was measured in day +12 3T3-L1s using ImageJ [37].

## 2-deoxy-glucose uptake assay

2-deoxy-glucose (2DOG) uptake assays were performed on mature adipocytes (day +8) following differentiation of MEFs in 24-well plates, according to standard protocols [38]. Briefly, MEFs were serum starved in Dulbecco's Modified Eagle's Medium (DMEM)/0.2% bovine serum albumin (BSA)/1% GlutaMax for 2 hours then washed three times in KRP buffer with 0.2% BSA. KRP buffer was prepared as: 0.6mM $Na_2HPO_4$, 0.4mM $NaH_2PO_4$, 120mM NaCl, 6 mM KCl, 1mM $CaCl_2$, 1.2mM $MgSO_4$ and 12.5mM HEPES adjusted to pH 7.4. At least one well on each plate served as a negative control by addition of 25μM cytochalasin B (Sigma Aldrich) to block all transporter-mediated glucose uptake. Cells were incubated with KRP/ 0.2% BSA ±100nM insulin for 20 minutes. After 15 minutes, 2-deoxyglucose (Sigma/Merck; radiolabelled 2DOG from PerkinElmer) was added to each well to a final concentration of 50μM and 0.25μCi/well. 5 minutes after addition of 2DOG, cells were quickly washed three

times in ice-cold PBS then lysed in 1% (v/v) Triton X-100. Uptake of 2DOG ($^3$H) was quantified using liquid scintillation counting and normalised to protein content.

For immunoblots of the insulin signalling cascade, MEFs were grown to day +10 of adipogenic differentiation, serum starved for 2 hours, then stimulated with 1nM insulin for 30 minutes. Cells were then washed in cold PBS and lysed in RIPA buffer (Sigma) with PhosSTOP Phosphatase Inhibitor (Roche) and Complete-Mini Protease Inhibitor (Roche), as described below. Lysates for immunoblotting of Glut1 and Glut4 were sonicated prior to quantification and were not denatured.

## RNAseq in knock-out MEFs

Bulk RNA sequencing (RNAseq) was performed on wild-type (WT) and *Mfn1*$^{-/-}$ MEFs at four time points: day -2 (pre-adipocyte), day 0 (initiation of adipogenic differentiation), day +3 (early adipocyte), and day +8 (mature adipocyte). RNAseq was performed on *Mfn2*$^{-/-}$ MEFs at two time points: day -2 (pre-adipocyte) and day +8 (mature adipocyte). Three independent biological replicates of each sample were used for analysis.

Cells were cultured and differentiated as described above, then briefly washed in PBS and lysed in RLT Buffer (Qiagen) with 1% 2-mercaptoethanol then passed through Qiashredder tubes (Qiagen). RNA was extracted using the RNeasy Isolation Kit (Qiagen) according to the manufacturers' protocol. RNA was quantified using Agilent 2100 Bioanalyzer (Agilent Technologies Inc) and only samples with RNA Integrity Number ≥8 were used for library preparation. cDNA libraries were made using Illumina TruSeq RNA sample kits and sequencing was performed on Illumina NovaSeq 6000 with paired-end 150bp reads (Novogene, Cambridge, UK). Raw reads all passed quality control for Qscore, error rate distribution, and AT/GC distribution.

Adapter sequences were removed from raw FASTQ files using cutadapt [39] and aligned to *Mus musculus* reference genome (GRCm38) using STAR [40]. Binary alignment/map (BAM) files were sorted using samtools [41] and counts were performed using featureCounts [42]. Differential gene expression (DGE) was performed using DESeq2 [43], where significance was considered as a Benjamini-Hochberg false-discovery rate (FDR) corrected p-value < .01.

Pathway analysis was performed with the EnrichR package for R [44–46] using significantly differentially expressed genes to determine enriched Hallmark gene sets [47]. Gene sets with FDR-corrected p-value < .05 were considered enriched. Figures were generated in R 4.0.2 [48] using packages pheatmap, ggplot2, and dplyr. RNAseq data is available in S1 Appendix and at Gene Expression Omnibus Study ID: GSE232502. All code used in analysis is available from: https://doi.org/10.5281/zenodo.5770057.

## Western blot studies

Cells were washed in cold PBS and lysed in RIPA buffer (Sigma) with PhosSTOP Phosphatase Inhibitor (Roche) and Complete-Mini Protease Inhibitor (Roche). Lysates were spun at 13,000rpm at for 15 minutes at 4°C then protein concentration quantified using BioRad DC Protein Assay Kit. 30–45µg protein lysates were mixed with NuPAGE 4x LDS buffer (Thermo-Fisher Scientific), containing 0.05% 2-mercaptoethanol, and denatured for 10 minutes at 95°C. Samples were run on 4–12% Bis-Tris gels (Invitrogen) and transferred onto a nitrocellulose membrane using iBlot-2 (ThermoFisher Scientific). Membranes were washed in Tris-buffered saline with 0.1% (vol/vol) Tween 20 (TBST, Sigma) before blocking in 5% (wt/vol) skimmed milk powder dissolved in TBST. Membranes were incubated with primary antibodies (S2 Table) at 4°C for 16 hours. Membranes were then washed with TBST five times for 5 minutes, followed by incubation with horseradish peroxidase (HRP)-conjugated secondary

antibodies for 1 hour at room temperature. Blots were developed using Immobilon Western Chemiluminescent HRP Substrate (Millipore) with images acquired on BioRad ChemiDoc Imaging system or GE ImageQuant LAS 4000.

## Statistical analysis

Continuous data were expressed as mean ± standard deviation. Normally distributed data were analysed by t-test (for two group pairwise comparison) and one-way ANOVA (for three or more groups) with post-hoc Bonferroni multiple comparisons test, where FDR-corrected p-value <0.05 was considered significant. Details of specific analyses and the number of replicates performed are reported in figure legends. All experiments were conducted at least three times using, where possible, randomisation of sample order and blinding of experimenters handling samples. Data were analysed using R 4.0.2 [48] and GraphPad Prism version 9 (GraphPad, San Diego).

## Results

### Characterization of undifferentiated knock-out MEFs

To assess the role of mitofusins in adipogenesis we utilised MEF lines deficient for *Mfn1* (*Mfn1*$^{-/-}$), *Mfn2* (*Mfn2*$^{-/-}$), or both (*Mfn1*$^{-/-}$*2*$^{-/-}$) (Fig 1A). We initially also characterised a line deficient for *Opa1* (*Opa1*$^{-/-}$), the GTPase that mediates fusion of the inner mitochondrial membrane downstream of mitofusin engagement [49, 50]. *Opa1*$^{-/-}$ MEFs are expected to have a specific defect in mitochondrial fusion.

We first assessed expression of core proteins involved in mitochondrial fusion and fission in undifferentiated MEFs. Loss of *Mfn1* increased expression of Mfn2, whilst loss of *Mfn2* or *Opa1* reduced expression of Mfn1 (Fig 1A & S1 Fig). Loss of each of the mitofusins was associated with no change in Opa1 expression whereas loss of both mitofusins led to a substantial reduction in Opa1 expression. Drp1, the main regulator of mitochondrial division [51], was expressed at similar levels in wild-type (WT) and *Mfn1*$^{-/-}$, but increased in *Mfn2*$^{-/-}$ MEFs and reduced in *Mfn1*$^{-/-}$*2*$^{-/-}$ and *Opa1*$^{-/-}$ MEFs. Fis1, a protein involved in, but not required for, mitochondrial fission [52, 53], was expressed at higher levels in *Mfn2*$^{-/-}$ MEFs but at similar levels in the other MEF lines.

As previously reported [14], *Mfn1*$^{-/-}$, *Mfn2*$^{-/-}$ and *Mfn1*$^{-/-}$*2*$^{-/-}$ MEFs exhibited fragmented mitochondria when compared to WT controls using either confocal- (Fig 1B) or transmission electron- microscopy (TEM) (Fig 1C). Analysis of the TEM images revealed more circular mitochondria (reduced length/width) in *Mfn1*$^{-/-}$ and *Mfn2*$^{-/-}$ MEFs (Fig 1D) without any difference in the mean mitochondrial perimeter (Fig 1E). Mitochondrial network disruption was most striking in *Mfn1*$^{-/-}$*2*$^{-/-}$ MEFs, which also consistently multiplied very slowly and failed to show any signs of adipocyte differentiation. Growth of *Opa1*$^{-/-}$ MEFs, which are also known to manifest severely disrupted mitochondrial morphology [54, 55], was similarly delayed and they too failed to show any signs of adipocyte differentiation, so these cells were not studied further in terms of adipogenesis. Loss of *Opa1* or both mitofusins has previously been found to prevent cells from responding to changes in metabolic demand [56] as well as manifesting low growth rates [49], which would limit the proliferation necessary for in vitro adipogenesis [57].

In the *Mfn1*$^{-/-}$*2*$^{-/-}$ and *Opa1*$^{-/-}$ MEFs we observed substantially reduced expression of mitochondrial oxidative phosphorylation complexes II, III, and IV (Fig 1F & S1 Fig). In contrast, Oxphos subunit expression was similar to WT cells in *Mfn1*$^{-/-}$ MEFs, except for slightly increased complex III expression. TOMM20 expression was increased in *Mfn1*$^{-/-}$ MEFs (Fig 1F & S1 Fig), suggesting increased mitochondrial mass. In *Mfn2*$^{-/-}$ cells expression of complexes II-V appeared normal whereas complex I expression was reduced (Fig 1F).

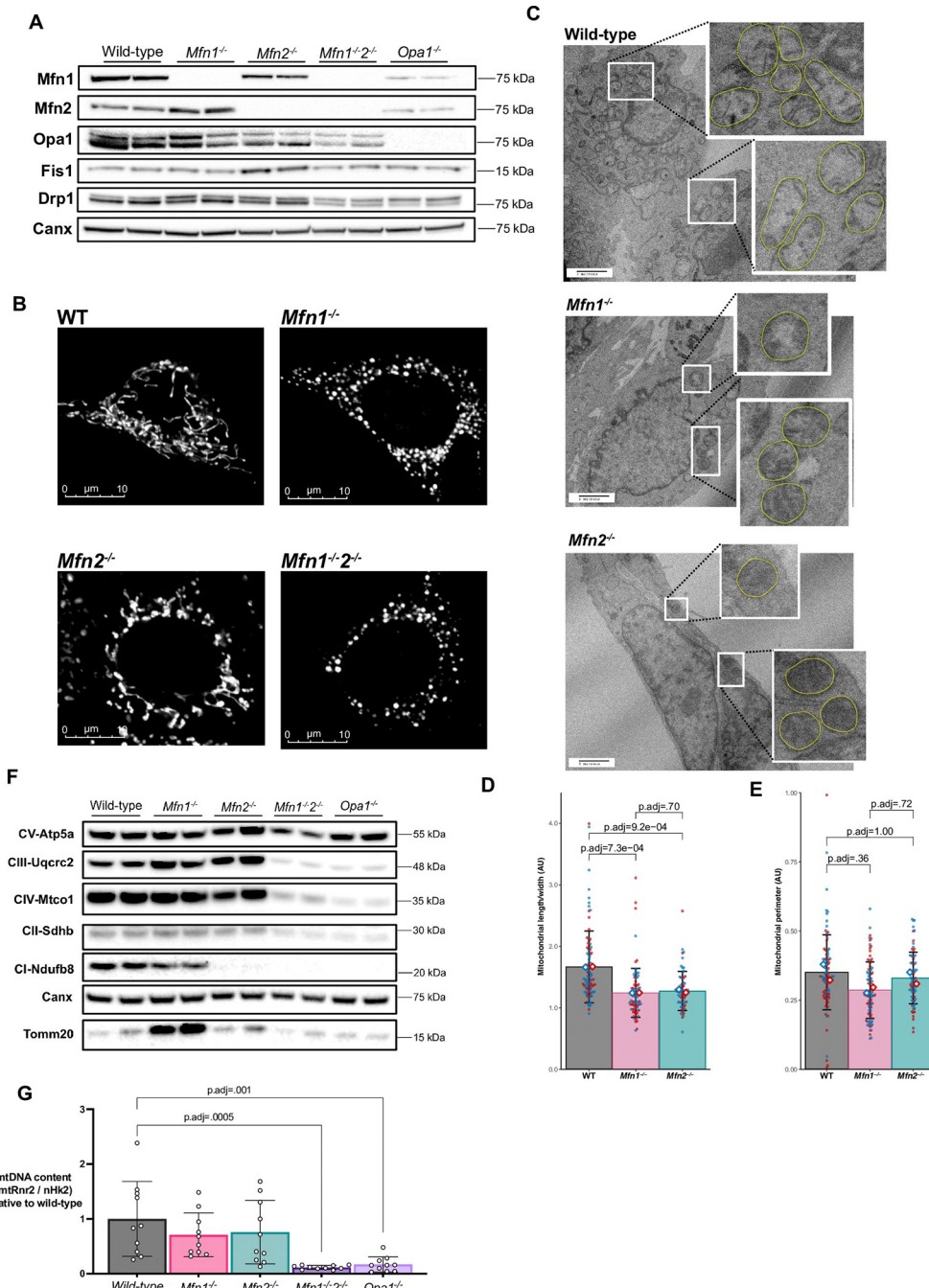

**Fig 1. Loss of mitofusins induces mitochondrial network fragmentation in mouse embryonic fibroblasts (MEFs).**
MEFs deficient for the mitochondrial fusion proteins *Mfn1*, *Mfn2*, both *Mfn1* and *Mfn2* (*Mfn1*⁻ᐟ⁻ *2*⁻ᐟ⁻), or *Opa1*⁻ᐟ⁻ were studied in their undifferentiated state. (A) Western blot confirming loss of protein from knock-out MEFs with calnexin (Canx) as loading control. (B) Mitotracker Orange imaging of MEFs on confocal microscopy demonstrated mitochondrial network fragmentation in knockout cell lines. (C) Transmission electron microscope (TEM) images of MEFs with zoomed-in images of mitochondria (highlighted in yellow). (D) Quantification of mitochondrial circularity (length/width) from TEM. Each red/blue dot represents one mitochondrion from two biological replicates. (E) Quantification of mitochondrial perimeter from TEM. Each red/blue dot represents one mitochondrion from two biological replicates. (F) Western blot of mitochondrial Oxphos complex subunits. (G) Mitochondrial DNA content expressed as mtRnr2 / nuclear Hk2 DNA from quantitative PCR. Each data point represents a separate biological replicate. All p-values represent pairwise comparisons between knock-outs and wild-type using t-tests, adjusted for multiple testing (p.adj). Data is representative of at least 3 independent replicates.

Mitochondrial DNA (mtDNA) is prone to depletion in cells with impaired mitochondrial fusion [58]. We found that double knock-out *Mfn1*$^{-/-}$*2*$^{-/-}$ and *Opa1*$^{-/-}$ MEFs had reduced mtDNA levels whereas loss of *Mfn1* or *Mfn2* in isolation did not significantly decrease mtDNA content (Fig 1G).

## Adipogenic differentiation of knock-out MEFs

We next assessed the ability of mitofusin KO MEFs to differentiate into adipocytes using a well-established adipogenic protocol. Surprisingly, we observed significantly increased lipid accumulation in *Mfn1*$^{-/-}$ MEFs, but not in *Mfn2*$^{-/-}$ MEFs, from day 4 of differentiation (Fig 2A), with increased neutral lipid content on both days 4 and 8 (Fig 2). A panel of mature adipocyte proteins, namely Plin1, Pparg, Fabp4, adiponectin and Glut4 (Fig 2D & S2 Fig) showed concomitant increases in expression. Expression of the insulin receptor (Insr) was also increased in these cells (Fig 2D & S2 Fig). Loss of *Mfn1* was associated with a greater increase in Pparg1 than Pparg2 (Fig 2D & S2 Fig). These expression changes were associated with increased insulin-stimulated Akt phosphorylation (Fig 2D & S2 Fig) and elevated insulin-stimulated 2-deoxy-glucose uptake compared to WT MEFs (p = 1.1x10$^{-12}$, Fig 2C). These findings suggest *bona fide* enhanced adipogenic differentiation rather than simple lipid accumulation in *Mfn1*$^{-/-}$ MEFs. In *Mfn2*$^{-/-}$ MEFs, adipogenic markers were generally reduced (Fig 2D & S2 Fig) and the impact of insulin on glucose uptake was decreased (Fig 2C). However, these cells did manifest high basal glucose uptake (WT 2.8 ± .6 vs. *Mfn2*$^{-/-}$ 11.7 ± 0.9 nmol/mg/min, p = 4.8x10$^{-13}$). We hypothesized that this might reflect increased Glut1 expression, but this was not the case (Fig 2D & S2 Fig), so at this stage we do not have a definitive explanation for this observation.

## Differentiation of knock-out MEFs over-expressing Pparg

MEFs are not fully committed preadipocytes, and show a relatively low rate of adipocyte differentiation in response to hormonal stimuli alone [59]. We thus next assessed whether over-expression of *Pparg2*, the master transcriptional driver of adipogenesis, would modify the impact of mitofusins deficiency on adipogenesis. Pparg2 was retrovirally overexpressed in WT and different mitofusins KO MEFs, with overexpression higher in *Mfn2*$^{-/-}$ MEFs than in WT and *Mfn1*$^{-/-}$ MEFs (S3A, S3B Fig). Upon adipogenic differentiation, we observed results consistent with those in untransduced cells, namely enhanced lipid accumulation in *Mfn1*$^{-/-}$ MEFs but similar lipid accumulation in the WT and *Mfn2*$^{-/-}$ MEFs (S3C & S3D Fig). There was increased expression of *Pparg1*, *Fabp4* and *Glut4* in *Mfn1*$^{-/-}$ MEFs with reduced expression of *Glut4* in *Mfn2*$^{-/-}$ MEFs despite higher baseline expression of *Pparg2* (S3E & S4 Figs). Expression of Plin1 was higher in both *Mfn1*$^{-/-}$ and *Mfn2*$^{-/-}$ MEFs compared to differentiated WT MEFs, demonstrating that loss of *Mfn1*, but not *Mfn2*, was associated with enhanced differentiation.

## Transcriptomic profiling of knock-out MEFs

To investigate how *Mfn1* deficiency enhances adipogenic differentiation we next performed bulk RNA sequencing of WT, *Mfn1*$^{-/-}$, and *Mfn2*$^{-/-}$ MEFs in undifferentiated (day -2) and differentiated (day +8) states. *Mfn1* and *Mfn2* null pre-adipocytes (day -2) both demonstrated significant differences to WT cells, and differences from each other (Fig 3A & 3B). Most strikingly, gene set enrichment analysis (GSEA) showed that the most highly enriched pathway in *Mfn1*$^{-/-}$ MEFs was 'hypoxia-related genes' (Fig 3C), whereas this gene set was downregulated in *Mfn2*$^{-/-}$ MEFs (Fig 3C). This gene set includes *Vegfa* as well as genes involved in the response to oxidative stress, such as *Selenbp1* and *Ankzf1*. Both knock-out lines showed

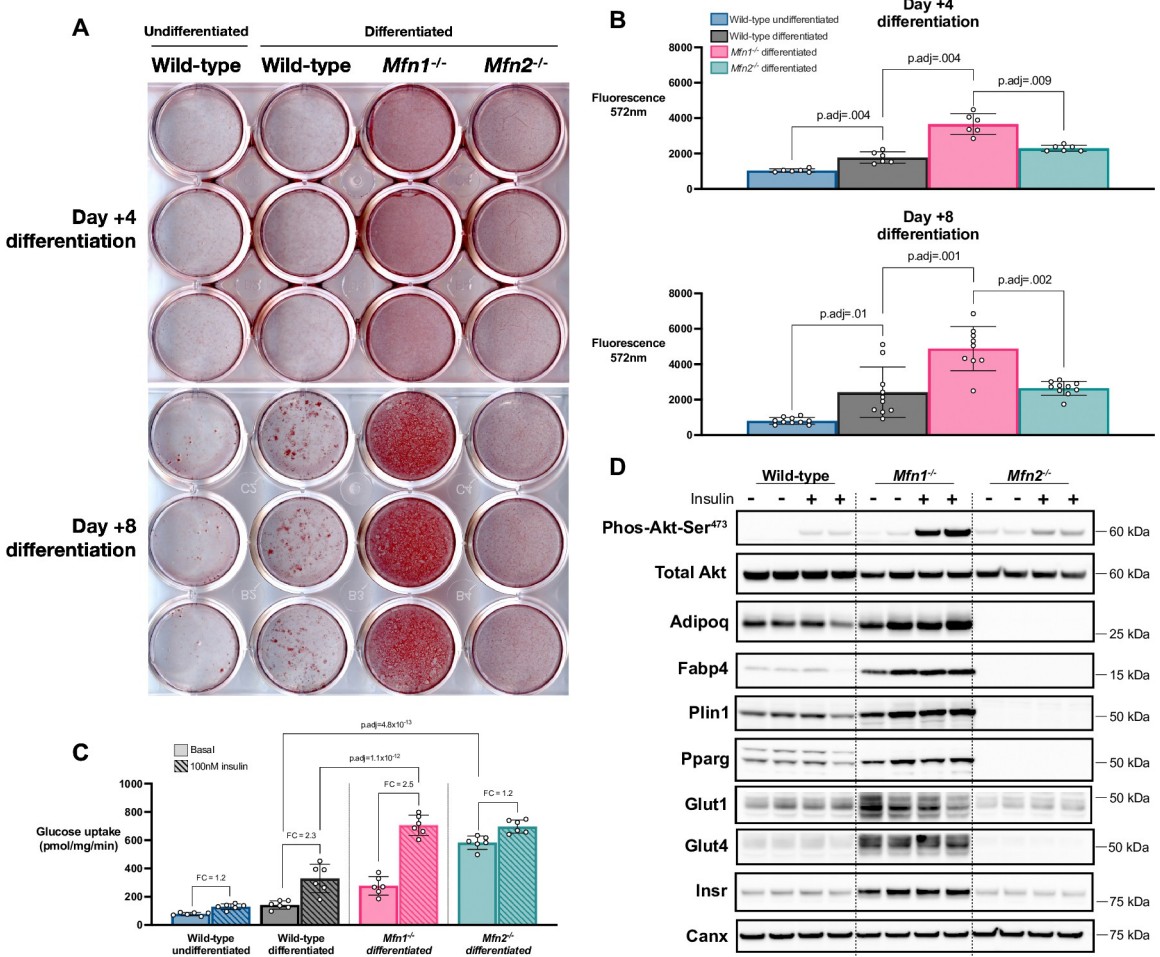

**Fig 2. *Mfn1*⁻/⁻ MEFs demonstrate enhanced adipogenesis.** MEFs underwent adipogenic differentiation using standard protocols and were assessed at day +4 and +8 of differentiation. (A) Oil Red O staining of differentiated MEFs compared to undifferentiated wild-type MEFs. (B) Fluorometric quantification of AdipoRed neutral lipid dye at day +4 & day +8. Each data point represents a separate biological experiment. (C) Glucose uptake assay (2-deoxy-D-glucose) at day +8 under basal and 100nM insulin conditions. Each data point represents a separate biological experiment. FC = Fold increase from basal to insulin-stimulated glucose uptake. (D) Western blot for components of the insulin signalling cascade and markers of adipocyte differentiation at day +10 with and without insulin stimulation. Calnexin (Canx) used as loading control. All p-values represent pairwise comparisons between knock-outs and wild-type using T-tests, adjusted for multiple testing (p.adj). Data is representative of at least 3 independent replicates.

enrichment of growth-related pathways (e.g. Estrogen Response Early), which was most evident in the *Mfn1*⁻/⁻ MEFs (Fig 3C). Genes implicated in the G2-M checkpoint gene set were upregulated in *Mfn2*⁻/⁻ MEFs but downregulated in *Mfn1*⁻/⁻ MEFs. *Mfn2*⁻/⁻ MEFs also showed upregulation of other pathways related to growth, including 'Apoptosis', 'Mitotic spindle', and 'E2F targets' (Fig 3C), consistent with the rapid growth of this cell line [60].

Although this analysis did not highlight the 'adipogenesis pathway', it did show that mRNA expression of some pro-adipogenic transcription factors, including *Pparg*, *Cebpa*, and *Zfp467* [61] was increased and that expression of *Zfp521* (coloured red in Fig 3A), an anti-adipogenic factor [62, 63], was reduced in *Mfn1*⁻/⁻ cells compared to WT(Fig 3D). These differences were sustained throughout the differentiation time-course. Despite the higher basal glucose uptake in *Mfn2*⁻/⁻ MEFs compared to WT (Fig 2D), there was no significant difference in *Slc2a1* (gene encoding Glut1): 0.2 log2FC, q-value = .03. Unlike models of *Mfn2* deletion *in vivo* [11], there

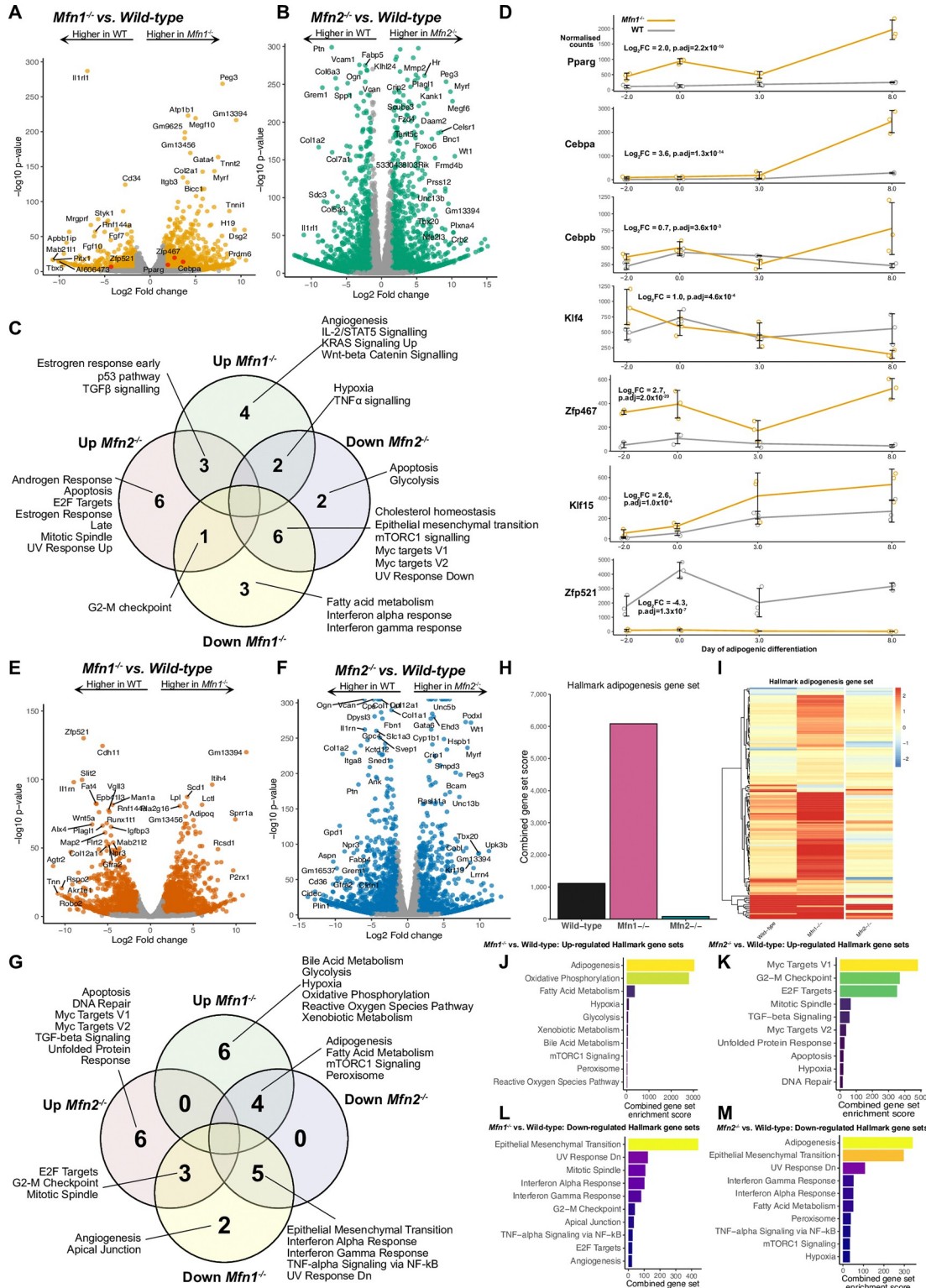

**Fig 3. *Mfn1*[-/-] MEFs demonstrate a pro-adipogenic transcriptional signature even in an undifferentiated state.** Bulk RNA sequencing was performed on wild-type (WT), *Mfn1*[-/-], and *Mfn2*[-/-] at day -2, 0, +3, and +8 of differentiation. (A) Volcano plot demonstrating significant differential gene expression (DGE) in *Mfn1*[-/-] MEFs vs. WT at day-2. Data from n = 3 biological replicates. All genes in orange show significant DGE (adjusted p-value <0.001 and log$_2$ fold change (log$_2$FC) greater or less than 1.5. Adipogenic transcription factors are shown in red. (B) Volcano plot for *Mfn2*[-/-] MEFs vs. WT at day -2. Genes in green show

significant DGE. (C) Venn diagram illustrating the significantly upregulated and downregulated Hallmark gene sets on transcriptomic analysis of *Mfn1*$^{-/-}$ and *Mfn2*$^{-/-}$ MEFs versus wild-type MEFs at day -2. (D) Time course from RNAseq demonstrating the change in seven key adipogenic transcription factors in WT and *Mfn1*$^{-/-}$ MEFs during differentiation. Data represents normalised counts per million. Log$_2$FC and adjusted p-values (p.adj) are derived from calculations of DGE at day -2, as described above. (E) Volcano plot for *Mfn1*$^{-/-}$ MEFs vs. WT at day+8. Data from n = 3 biological replicates. All genes in red show significant DGE. (F) Volcano plot for *Mfn2*$^{-/-}$ MEFs vs. WT at day +8. Genes in blue show significant DGE. (G) Significantly upregulated and downregulated Hallmark gene sets from *Mfn1*$^{-/-}$ and *Mfn2*$^{-/-}$ MEFs versus wild-type MEFs at day +8. (H) Hallmark adipogenesis gene set enrichment score from pathway analysis following DGE comparison of day -2 versus day +8 for each cell line. (I) Heatmap illustrating log2fold change for all the 200 genes included in the Hallmark adipogenesis gene set (data from (H)). Pathway analysis showing up-regulated (J-K) and down-regulated (L-M) pathways for *Mfn1*$^{-/-}$ and *Mfn2*$^{-/-}$ MEFs versus WT at day +8.

was no evidence of activation of the integrated stress response in *Mfn2*$^{-/-}$ MEFs in the undifferentiated state (Fig 3C).

RNA sequencing of mature (day +8) adipocytes (Fig 3E, 3F) demonstrated expression profiles consistent with enhanced differentiation in *Mfn1*$^{-/-}$ MEFs and reduced differentiation in *Mfn2*$^{-/-}$ MEFs. For example, relative to WT, *Mfn1*$^{-/-}$ MEFs showed increased expression of *Plin1*, *Pparg*, *Slc2a4* (encoding Glut4), *Fabp4*, and *Cd36*, whilst each of these were reduced in *Mfn2*$^{-/-}$ MEFs. This was also observed on gene set enrichment analysis using Hallmark gene sets [47]: when comparing day -2 pre-adipocytes and day +8 mature adipocytes within each cell line, *Mfn1*$^{-/-}$ had markedly greater enrichment of the adipogenesis gene set (Fig 3H–3I). Similarly, pathway analysis comparing day +8 WT and KO cell lines revealed upregulation of the adipogenesis gene set for *Mfn1*$^{-/-}$ MEFs and downregulation of the same gene set in *Mfn2*$^{-/-}$ MEFs (Fig 3G–3M). *Mfn1*$^{-/-}$ MEFs also manifested upregulation of oxidative phosphorylation, hypoxia, and reactive oxygen species pathway gene sets (Fig 3J). Thus, the transcriptomic data supports a pro-adipogenic phenotype in *Mfn1*$^{-/-}$, but not *Mfn2*$^{-/-}$, MEFs.

## siRNA knockdown and differentiation of 3T3-L1 preadipocytes

Given the surprisingly divergent effects of Mfn1 and Mfn2 deficiencies on MEF adipogenesis, we next sought to corroborate findings in an independent cellular model. We thus evaluated the impact of siRNA mediated *Mfn1* and *Mfn2* knockdown in 3T3-L1 pre-adipocytes. The 3T3-L1 line is a MEF subclone first established in the 1970s based on its high propensity for adipogenic differentiation, and has since been widely used as a model of *in vitro* adipogenesis [64, 65]. Treating 3T3-L1s with siRNAs targeting *Mfn1* or *Mfn2* effectively reduced target protein expression by more than 95% (Fig 4A and S5A, S5B Fig). Like the KO cell lines, knock-down of Mfn2 reduced Mfn1 expression, whereas knock-down of Mfn1 modestly increased Mfn2 expression (Fig 4A & S5 Fig).

Throughout differentiation of wild-type 3T3-L1s there was an increase in expression of both markers of mitochondrial fusion (Mfn1 & Mfn2) and fission (Fis1, S6A Fig). In keeping with the MEF data, 3T3-L1s treated with *Mfn1*-targeted siRNA demonstrated increased lipid accumulation (Fig 4B, 4C). This observation was broadly replicated when the two anti-*Mfn1* siRNAs were used independently, such that Mfn1 knock-down efficiency (S7A Fig) correlated with *Pparg* expression (S7B Fig) and degree of lipid accumulation (S7C, S7D Fig). Whilst there was no difference in total lipid accumulation between scrambled and *Mfn2* knockdown cells, lipid droplet size was increased in the Mfn2 knockdown cells (Fig 4B & S6B Fig), as has been previously reported in knock-out MEFs [66]. There was no difference in lipid droplet size between scrambled and *Mfn1* knock-down cells (Fig 4B & S6B Fig).

Expression of Pparg1 and Glut4 protein was increased in *Mfn1* knockdown cells (Fig 4A & S5 Fig), whereas expression of Plin1 and Fabp4 were similar to that of the control cells. *Mfn2* knock-down reduced lipid accumulation and expression of Plin1, Fabp4, Pparg1, Pparg2, and Glut4 proteins (Fig 4 & S5 Fig).

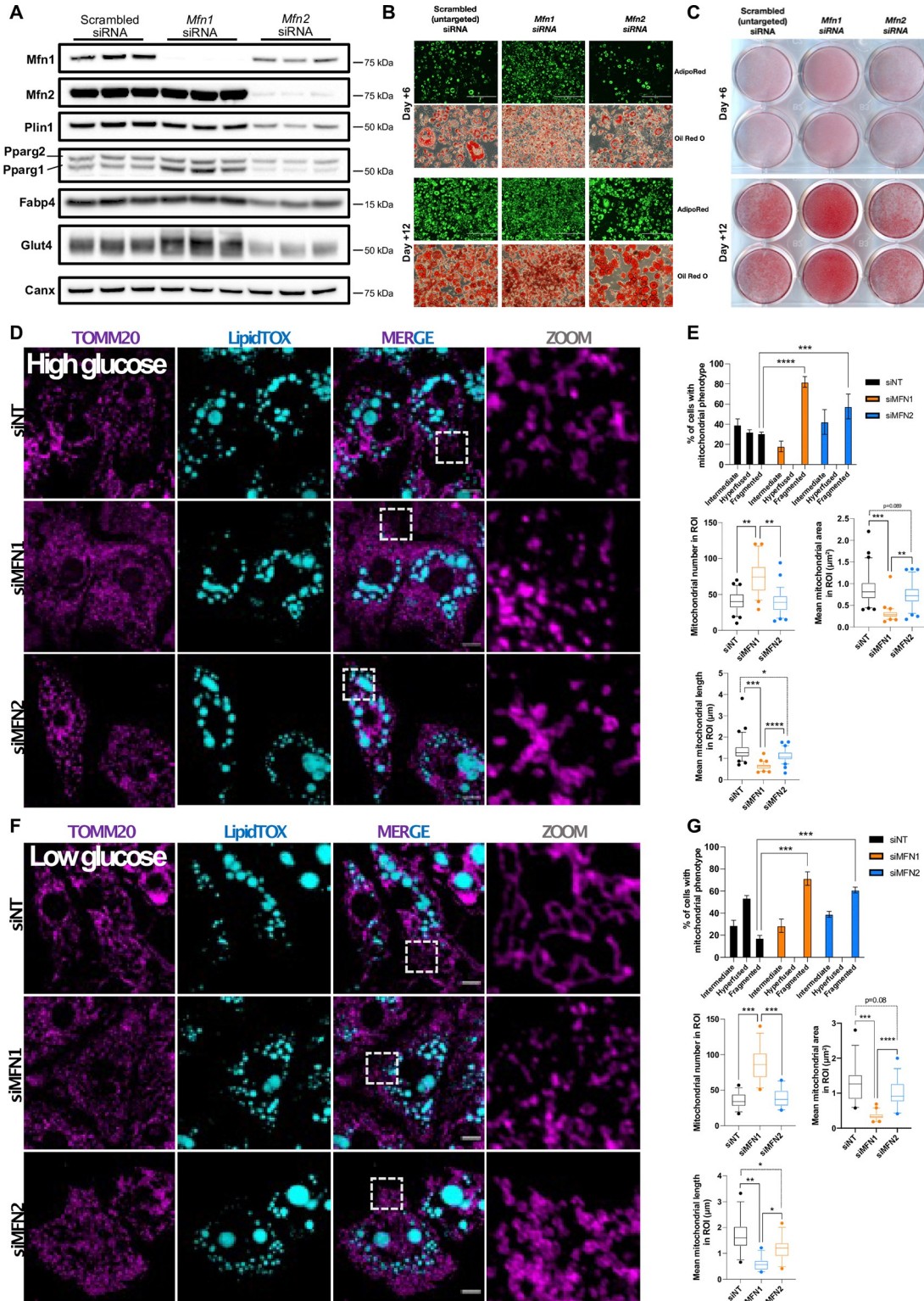

**Fig 4. siRNA knockdown of *Mfn1* in 3T3-L1s enhances adipogenesis.** 3T3-L1s were treated with scrambled, *Mfn1*-targeted, or *Mfn2*-targeted siRNA on alternate days from day -2 to day +12 of differentiation and assessed on days +6 and +12. (A) Western blot from 3T3-L1 at day +12 showing efficacy of *Mfn1/Mfn2* knock-down and expression of selected adipogenic genes. Data represents three biological replicates. (B) Light microscope images of AdipoRed fluorescence (green = lipid) and Oil Red O stained 3T3-L1s at day +6 and +12 of differentiation. (C) Oil Red O staining of 3T3-L1s at day +6 and +12 of differentiation. (D)

Effect of siRNA knock-down on mitochondrial morphology assessed by confocal microscopy in 3T3-L1s cells at day +12 of differentiation under high-glucose conditions. Mitochondria were labelled using an anti-TOMM20 antibody (purple) and neutral lipids were stained with LipidTox (blue). Scale bars: 10 μm. (E) Quantification of mitochondrial morphology, and different mitochondrial parameters including number, area, and length in 225 μm$^2$ region of interests (ROI) from (D). (F) Effect of siRNA knock-down on mitochondrial morphology assessed by confocal microscopy in 3T3-L1s cells at day +12 of differentiation under low glucose conditions. Mitochondria were labelled using an anti-TOMM20 antibody (purple) and neutral lipids were stained with LipidTox (blue). Scale bars: 10 μm. (G) Quantification of mitochondrial morphology, and different mitochondrial parameters including number, area, and length in 225 μm$^2$ ROIs from (F). All p-values represent pairwise comparisons between knock-outs and wild-type using t-tests, adjusted for multiple testing (p.adj). Data is representative of at least 3 independent replicates.

Mitochondrial morphology analysis revealed that loss of Mfn1 and Mfn2 both led to mitochondrial fragmentation, with Mfn1 silencing inducing more drastic changes of the mitochondrial network in mature (day +12) differentiated adipocytes (Fig 4D, 4E), even though, as previously reported [25], the mitochondrial network of wild-type 3T3-L1s manifests increased fragmentation in late differentiation (Fig 4D, 4E). Nutritional deprivation is known to induce mitochondrial fusion in several different cell lines [67]. In keeping with this, incubating the 3T3-L1 adipocytes in low glucose media or without serum increased fusion of the mitochondrial network in siNT treated control cells characterized by mitochondrial elongation (S6 Fig), and accentuating the impact of Mfn1 and Mfn2 deficiencies on the mitochondrial network (Fig 4F, 4G). These findings in 3T3-L1 adipocytes support the observation that loss of mitofusins 1 and 2 have specific effects on adipogenic potential despite both causing mitochondrial network fragmentation.

### Effect of ROS scavenging on MEF differentiation

Prior work has suggested that reactive oxygen species (ROS) promote adipogenesis [68, 69]. To test the hypothesis that altered ROS production might be mediating the enhanced adipogenesis observed in *Mfn1*$^{-/-}$ MEFs, we added a ROS-scavenger (N-acetylcysteine (NAC)) during adipogenic differentiation of the knock-out cell lines. Consistent with previous reports [68, 69], NAC inhibited lipid accumulation and expression of adipogenic markers in wild-type MEFs (Fig 5). However, i n *Mfn1*$^{-/-}$ MEFs, addition of NAC had very little impact on either lipid accumulation (Fig 5A) or adipocyte protein expression (Fig 5B) expression. Similarly, NAC had no effect on adipogenic differentiation of *Mfn2*$^{-/-}$ MEFs.

### Discussion

Prompted by observations of a human monogenic disease, which highlighted the importance of the mitochondrial network in adipose tissue, we sought to investigate the roles of two mitochondrial fusion proteins in adipogenesis. Although human genetics has established that a specific genetic mutation of *MFN2* has a profound impact on white adipose tissue development and function, our analysis of adipogenesis in cultured cells suggested that deletion or knockdown of *Mfn1* enhances adipogenesis whereas *Mfn2* deletion or knockdown tended to impair differentiation.

The absence of Mfn2 in MEFs subtly reduced adipogenesis, indicated by reduced expression of Plin1, Pparg, Fabp4, and Glut4, with no change in neutral lipid accumulation. This could be due to differences in gene expression (as found in RNAseq analysis) and/or changes in lipid metabolism (synthesis, intracellular trafficking and/or oxidation), leading to similar lipid accumulation despite lower adipogenic capacity. *Mfn2*$^{-/-}$ adipocytes also manifested increased basal glucose uptake, which may relate to impaired mitochondrial bioenergetics causing increased glycolysis, as has been observed in brown adipocytes lacking mitofusin 2

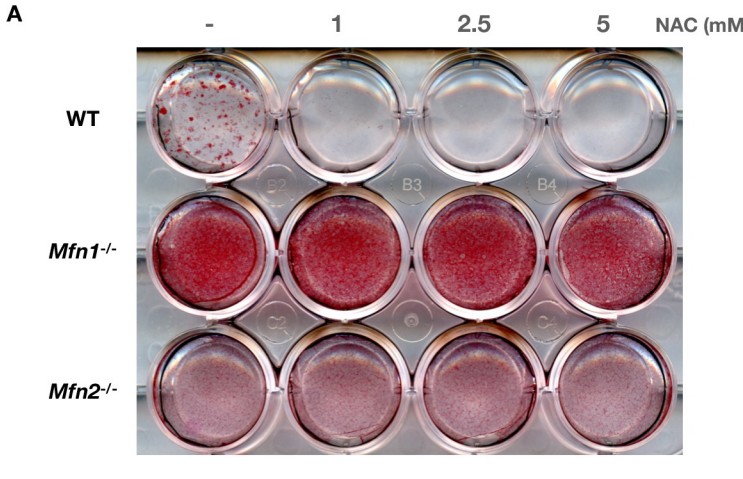

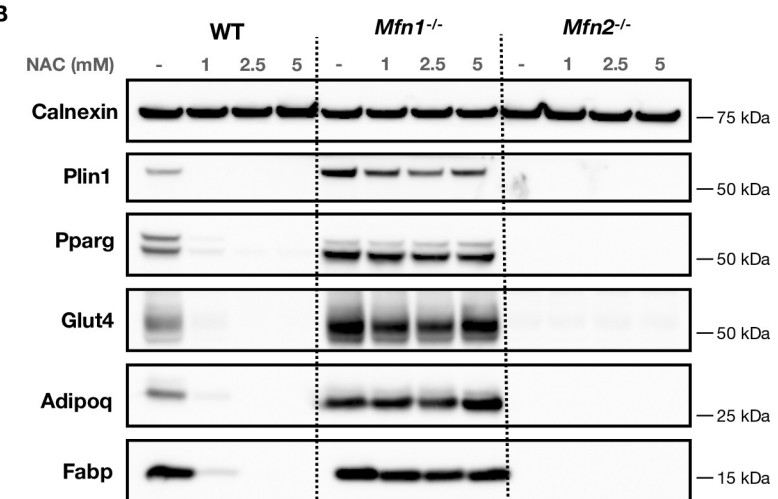

**Fig 5. Reactive oxygen species scavenger inhibits differentiation in wild-type MEFs.** Wild-type (WT), *Mfn1⁻/⁻*, and *Mfn2⁻/⁻* MEFs were treated with N-acetylcysteine (NAC) at 1mM, 2.5mM, and 5mM on alternate days from day -2 to day +8 of differentiation. (A) Oil Red O staining of MEFs at day +8 of differentiation. (B) Western blot from MEFs at day +8 showing expression of selected adipocyte proteins. Data is representative of at least 3 independent replicates.

[21]. The mechanism underlying this is unclear, particularly given no elevation in Glut1 expression in these cells. Akt 2 phosphorylation did not significantly increase in *Mfn2⁻/⁻* MEFs in response to insulin and there was also very low levels of GLUT4 expression in these cells. Data from tissue-specific knock-out mice suggests that loss of *Mfn2* can perturb glucose homeostasis. Specifically, deletion of *Mfn2* in all adipocytes [20] or in brown adipocytes alone [21] caused reduced expression of multiple oxidative phosphorylation subunits and impaired cold tolerance yet, paradoxically, both mouse lines were protected from systemic insulin resistance. Mice in which Mfn2 was inducibly deleted in all adipocytes in adulthood showed increased obesity and elevated blood glucose [22]. Further work will be required to comprehensively explain these somewhat discordant data.

In both *Mfn1⁻/⁻* knock-out MEFs and *Mfn1* knock-down 3T3-L1s there was a significant increase in lipid accumulation. In our view, this reflects enhanced adipogenesis, as it was associated with elevated expression of several stereotypical adipogenic markers (e.g. Plin1, FABP4 and *Glut4*). *Mfn1⁻/⁻* MEFs had increased Pparg expression even in the undifferentiated state, and RNA sequencing showed that expression of some other pro-adipogenic transcription

factors (*Cebpa*, *Cebpb*, *Klf4*, *Klf15*, and *Zfp467*) was increased, suggesting that Mfn1 deficiency may prime cells transcriptionally to favour adipogenesis. Indeed, the changes in gene expression observed during differentiation showed a stronger signature for adipogenic differentiation in *Mfn1*[-/-] than wild-type MEFs.

The exact mechanism behind this is currently unclear. One potential explanation could involve the generation of oxidised lipids secondary to ROS generation from altered mitochondrial function. This is a plausible hypothesis for these observations as increased ROS early in adipogenesis has been reported to accelerate differentiation [68, 69]. In addition, humans with loss of function of selenoproteins have chronically elevated oxidative stress and have increased adiposity yet are more insulin sensitive [70]. Use of NAC as a ROS scavenger had minimal impact on lipid accumulation or protein expression in *Mfn1*[-/-] MEFs. Multiple tissue-specific knock-outs for both *Mfn1* [71] and *Mfn2* [12, 72] have been reported to be associated with changes in ROS. Liver-specific knock-out of *Mfn2* was associated with increased hydrogen peroxide and reduced expression of subunits from Complex IV [12]. In addition, *Mfn2*[-/-] MEFs have reduced membrane potential compared to wild-type [73], which, along with lower Complex I expression, may influence their ROS generation through reverse electron transport [74, 75]. We have attempted to quantify ROS production in the adipocyte lines described herein, but have found it difficult to reliably quantify ROS in these cell types. The experiments we report using NAC could be interpreted in at least two ways in our view. Firstly, the lack of impact of NAC in *Mfn1* null MEFs may be because ROS are not involved in mediating the enhanced adipogenesis observed in these cells or, secondly, perhaps the change in ROS associated with NAC was 'overwhelmed' by the changes in ROS in this cell line, such that whereas this impaired adipogenesis in wildtype cells, it did not do so in the Mfn1[-/-] cells. Further work in this field should include robust measurement of ROS in pre- and mature-adipocytes, including mitofusin knock-outs, as well as use of alternative ROS scavengers & inducers (e.g. mitoTEMPO and mitoQ [76]).

It is unclear precisely how loss of the mitofusins alters metabolite substrate use in adipocytes derived from MEFs. In animals fed a high-fat diet, the presence of Mfn2 is needed to promote lipolysis and its absence (in brown adipose tissue) causes a shift towards glycolysis [20, 21, 77]. Bioenergetic studies (e.g. Seahorse analysis) in pre- and mature- adipocytes (including mitofusin knockouts) would help to delineate the effect of *Mfn1*/*Mfn2* loss on this phenotype.

In the current study we were unable to differentiate *Mfn1*[-/-]*2*[-/-] double knock-out MEFs into adipocytes, in contrast to earlier work by McFie *et al.* [66]. This may be due to differences in passage and acquired clonal changes in MEFs. Clonal changes could also account for differences observed in MEFs (given that they were derived from different mice). Whilst we have supporting evidence through knock-down in 3T3-L1 adipocytes, 'rescue' experiments in MEFs (e.g. re-expression of *Mfn1* in *Mfn1*-/- MEFs) could help to strengthen these observations.

An important limitation of this work is a lack of *in vivo* data to support this hypothesised role for *Mfn1* in adipogenesis. Adipose-specific *Mfn1* knock-out mice, although briefly said in one study not to have a gross adipose phenotype, have not been characterised in any detail [20]. An important limitation of such a model in any case relates to the fact that deletion of 'floxed' *Mfn1* was under control of the adiponectin-promoter [20]. Adiponectin is only expressed in mature adipocytes [78] so such an experiment does not test the impact of Mfn1 deficiency early in adipocyte development. The data reported herein was also exclusively conducted in murine cells. Whilst we found broadly concordant observations in two different cell lines we have not studied human adipocytes. There are currently no known human disorders linked to *MFN1* mutations but based on our findings, we would hypothesise that individuals carrying such mutations may be more insulin sensitive and therefore may not present with a

clinical disease phenotype. However, germline *Mfn1* mutations would also perturb Mfn1 function in other tissues which may have a greater phenotypic impact than that related to altered adipose expression. On the other hand, human population frequencies of sequence variants in *MFN1* and *MFN2* suggest that loss-of-function mutations in *MFN1* may not be selected against, unlike loss-of-function variants in *MFN2* [79].

Another line of investigation requiring further work is detailed characterisation of mitochondrial function in 3T3-L1 and MEFs during adipogenic differentiation. Such studies could include bioenergetic analyses, measurement of mitochondrial membrane potential, and isolation of mitochondria for high-resolution respirometry. These studies may provide additional mechanistic insights into the role of mitochondria in cultured adipocytes.

## Conclusions

At endogenous levels, MFN1 appears to exercise tonic physiological restraint on differentiation to adipocytes in culture, whereas the closely related MFN2 is necessary for expression of the full programme of adipogenesis.

## Supporting information

**S1 Fig. Quantification of protein expression from immunoblots in Fig 1.** Quantification of protein expression (relative to wild-type) for all proteins shown in Fig 1A and 1F. Each data point represents a separate biological replicate. Stars indicate p-values following pairwise comparisons between groups: * $p < 0.5$, ** $p < .01$, *** $p < .001$, **** $p < .0001$. ns, not significant.
(PDF)

**S2 Fig. Quantification of protein expression from immunoblots for insulin-stimulation of differentiated MEFs in Fig 2D.** Quantification of protein expression (relative to wild-type) for all proteins shown in Fig 2D from knock-out MEFs. Each data point represents a separate biological replicate. Stars indicate p-values following pairwise comparisons between groups: * $p < 0.5$, ** $p < .01$, *** $p < .001$, **** $p < .0001$. ns, not significant.
(PDF)

**S3 Fig. Increased adipogenesis in *Mfn1*$^{-/-}$ MEFs even in setting of *Pparg2* over-expression.** *Pparg2* was over-expressed in MEFs to enhance their adipogenic differentiation capacity and cells were assessed on day +4 and +8 of protocols. (A) Western blot illustrating the relative Pparg2 over-expression in cell lines used compared to non-transduced wild-type cells. (B) Quantification of Pparg2 expression from the Western blot in panel (A). (C) Oil Red O staining differentiated MEFs compared to undifferentiated wild-type MEFs. (D) Fluorometric quantification of AdipoRed neutral lipid dye at day +4 & day +8. Each data point represents a separate biological experiment. Data is expressed relative to undifferentiated wild-type for each biological replicate. (E) Western blot for markers of adipocyte differentiation at day +8 from MEFs over-expressing *Pparg2*. All p-values represent pairwise comparisons between knock-outs and wild-type using t-tests, adjusted for multiple testing (p.adj). Data is representative of at least 3 independent replicates.
(PDF)

**S4 Fig. Quantification of protein expression from immunoblots for differentiation of *Pparg2* over-expressing knock-out MEFs from S3E Fig.** (A-E) Quantification of protein expression from knock-out MEFs over-expressing Pparg2, from S3E Fig. Each data point represents a separate biological replicate. Stars indicate p-values following pairwise comparisons between groups: * $p < 0.5$, ** $p < .01$, *** $p < .001$, **** $p < .0001$. ns, not significant.
(PDF)

**S5 Fig. Quantification of protein expression from immunoblots in Fig 4A.** Quantification of protein expression (relative to scrambled control) for all proteins shown in Fig 4A. Each data point represents a separate biological replicate. Stars indicate p-values following pairwise comparisons between groups: * p<0.5, ** p < .01, *** p < .001, **** p < .0001. ns, not significant.
(PDF)

**S6 Fig. Low glucose treatment induces mitochondrial network fusion in differentiated mature 3T3-L1 adipocytes.** (A) Representative western blot showing change in mitofusin expression in 3T3-L1 cells during adipogenic differentiation. (B) Measurement of lipid droplet width from oil red O staining of day +12 3T3-L1 adipocytes treated with scrambled, *Mfn1*-targeted, or *Mfn2*-targeted siRNA on alternate days from day -2 to day +12 of differentiation. Data from four biological replicates. (C) Representative confocal images of 3T3-L1s cells at day +12 of differentiation under high glucose (HG) and low glucose (LG) conditions. Mitochondria were labelled using an anti-TOMM20 antibody (purple) and neutral lipids were stained with LipidTox (blue). Scale bars: 10 μm. (D) Quantification of mitochondrial morphology, and different mitochondrial parameters including number, area, and length in 225 μm² region of interests from (A). Stars indicate p-values following pairwise comparisons between groups: * p<0.5, *** p < .001.
(PDF)

**S7 Fig. Optimisation and efficacy of anti-*Mfn1* siRNAs on adipogenic differentiation of 3T3-L1s.** Undifferentiated 3T3-L1 pre-adipocytes were treated with three separate Invitrogen Silencer Select siRNAs (CatIDs: s85003 = 'siRNA03', s85002 = 'siRNA02', s85004 = 'siRNA01') for 24-hours, each siRNA was used at 10nmol/L, 30nmol/L, and 50nmol/L. A, Western blot demonstrating Mfn1 protein knock-down efficacy. Following this, 3T3-L1 adipocytes were treated with s85002 (= 'siRNA02') and s85004 (= 'siRNA01') until day +10 differentiation. B, Relative mRNA expression of markers of adipogenesis at day +10 differentiation. C, Lipid accumulation at day + 10 as measured by fluorescence using AdipoRed. D, Representative images for adipogenic differentiation using brightfield microscopy and AdipoRed staining. Data from n = 2 biological repeats.
(PDF)

**S1 Table. Reagents and cell lines.** List of all reagents and equipment used in this study.
(DOCX)

**S2 Table. Antibodies used.** List of all primary and secondary antibodies used in this study.
(DOCX)

**S1 Appendix. RNA sequencing counts tables.** Normalised counts per million per gene from RNA sequencing of MEFs. dm2, day -2 of differentiation; dm0, day zero of differentiation; dp3, day +3 of differentiation; dp8, day +8 of adipogenic differentiation; m1, Mfn1-/-; m2, Mfn2-/-; WT, wild-type MEFs.
(XLSX)

**S1 Raw images.**
(PDF)

## Acknowledgments

We are grateful for the support of the Cambridge Advanced Imaging Centre (University of Cambridge, UK) for performing, processing, and imaging of transmission electron microscope

studies. We also acknowledge the light microscopy of the Mitochondrial Biology Unit. We thank M. Mimmack for his technical assistance with cell culture and assays.

## Author Contributions

**Conceptualization:** Stephen O'Rahilly S., Julien Prudent, Robert K. Semple, David B. Savage.

**Data curation:** Jake P. Mann, Luis Carlos Tábara, Satish Patel, Pushpa Pushpa, Anna Alvarez-Guaita, Liang Dong, Afreen Haider, Koini Lim, Panna Tandon, Fabio Scurria, Daniel J. Fazakerley.

**Formal analysis:** Jake P. Mann, Luis Carlos Tábara, Satish Patel, Pushpa Pushpa, Anna Alvarez-Guaita, Liang Dong, Afreen Haider, Koini Lim, Panna Tandon, Fabio Scurria, Daniel J. Fazakerley.

**Funding acquisition:** Jake P. Mann, Luis Carlos Tábara, James E. N. Minchin, Stephen O'Rahilly S., Daniel J. Fazakerley, Julien Prudent, Robert K. Semple, David B. Savage.

**Investigation:** Jake P. Mann, Luis Carlos Tábara, Satish Patel, Pushpa Pushpa, Anna Alvarez-Guaita, Liang Dong, Afreen Haider, Koini Lim, Panna Tandon, Fabio Scurria, James E. N. Minchin, Stephen O'Rahilly S., Daniel J. Fazakerley, Julien Prudent, Robert K. Semple, David B. Savage.

**Methodology:** Jake P. Mann, Luis Carlos Tábara, Satish Patel, Pushpa Pushpa, Anna Alvarez-Guaita, Liang Dong, Afreen Haider, Koini Lim, Panna Tandon, Fabio Scurria, James E. N. Minchin, Stephen O'Rahilly S., Daniel J. Fazakerley, Robert K. Semple, David B. Savage.

**Project administration:** Jake P. Mann, Robert K. Semple, David B. Savage.

**Supervision:** James E. N. Minchin, Stephen O'Rahilly S., Daniel J. Fazakerley, Julien Prudent, Robert K. Semple, David B. Savage.

**Writing – original draft:** Jake P. Mann.

**Writing – review & editing:** Jake P. Mann, Luis Carlos Tábara, Satish Patel, Pushpa Pushpa, Anna Alvarez-Guaita, Liang Dong, Afreen Haider, Koini Lim, Panna Tandon, Fabio Scurria, James E. N. Minchin, Stephen O'Rahilly S., Daniel J. Fazakerley, Julien Prudent, Robert K. Semple, David B. Savage.

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
