## [Decision Letter · Decision Letter 0]

15 Aug 2023

PONE-D-23-15779Loss of Mfn1 but not Mfn2 enhances adipogenesisPLOS ONE

Dear Dr. Savage,

Thank you for submitting your manuscript to PLOS ONE. After careful consideration, we feel that it has merit but does not fully meet PLOS ONE’s publication criteria as it currently stands. Therefore, we invite you to submit a revised version of the manuscript that addresses the points raised during the review process.

Both reviewers are in favor of publishing the manuscript but have expressed concerns.

Reviewers have indicated both minor and major points and responding to them will  improve the manuscript.

We look forward to receiving your revised manuscript.

Kind regards,

Benedetta Ruzzenente

Academic Editor

PLOS ONE

“Wellcome Trust (WT):Jake P Mann 216329; Wellcome Trust (WT):Robert Kenneth Semple 210752; Wellcome Trust (WT):David B Savage 219417; Wellcome Trust (WT):Stephen O'Rahilly 214274; UKRI | Medical Research Council (MRC):Stephen O'Rahilly MRC_MC_UU_12012.1; UKRI | Medical Research Council (MRC):Julien Prudent MC_UU_00015/7; UKRI | Medical Research Council (MRC):Julien Prudent MC_UU_00028/5; UKRI | Biotechnology and Biological Sciences Research Council (BBSRC):Luis Carlos Tabara BB/W008467/1; UKRI | Medical Research Council (MRC):James EN Minchin MR/S025685/1; UKRI | Medical Research Council (MRC):Daniel J Fazakerley MR/S007091/1; UKRI | Medical Research Council (MRC):Stephen O'Rahilly MC_UU_00014/5; Wellcome Trust (WT):Stephen O'Rahilly 208363/Z/17/Z”

Additional Editor Comments:

Both reviewers are in favor of publishing the manuscript but have expressed concerns.

Reviewers have indicated both minor and major points and responding to them will improve the manuscript.

Reviewers' comments:

Reviewer's Responses to Questions

**Comments to the Author**

1. Is the manuscript technically sound, and do the data support the conclusions?

Reviewer #1: Partly

Reviewer #2: Yes

2. Has the statistical analysis been performed appropriately and rigorously? 

Reviewer #1: Yes

Reviewer #2: Yes

3. Have the authors made all data underlying the findings in their manuscript fully available?

Reviewer #1: Yes

Reviewer #2: Yes

4. Is the manuscript presented in an intelligible fashion and written in standard English?

Reviewer #1: Yes

Reviewer #2: Yes

5. Review Comments to the Author

Reviewer #1: This manuscript has been carefully reviewed by 3 reviewers, whose comments are well aligned. The authors performed experiments to address some of the reviewers’ comments, but many key questions remained unanswered.

1, Characterization of mitochondrial functions in 3T3-L1 and/or MEFs as suggested by the reviewers would provide significant mechanistic insights. The analysis was presented in many publications. However, little has been done to address the differential regulation of mitochondrial functions by Mfn 1 and 2.

2, Utilization of additional siRNA in 3T3-L1 cells and/or another line of Mfn1/2 KO MEF is recommended if re-expression of Mfn1/2 is not performed in KO MEFs as suggested by the reviewers.

Additional comments:

1, Mfn2-/- adipocytes have significant lower levels of GLUT1 and 4 than Mfn1-/- adipocytes (Fig 2D), but have higher basal rate of glucose uptake (Fig 2C). Please explain. Moreover, the insulin effect on glucose uptake is weak in the adipocytes, indicating that the differentiation procedure is not optimal.

2, Quantification of lipid accumulation of Mfn1-/- adipocytes (day 4 vs day 8) in fig 2B is inconsistent with the images in fig 2A. Combination of the two panels in fig 2B allows the comparison of lipid accumulation during adipogenesis.

3, Ln 151, concentrations of the stock solutions (sodium pyruvate, non-essential amino acids, penicillin/streptomycin) should be provided.

Reviewer #2: This study showed that loss of Mfn1 increases adipogenesis, while loss of Mfn2 decreases the expression of adipogenic markers without notable differences in lipid droplet. The authors suggest reactive oxygen species (ROS) as a potential mediator of the increased differentiation observed in Mfn1-/- cells. A role for ROS in stimulating adipogenesis was recently demonstrated by Singh et al. (Antioxidants and Redox Signaling, 2023) and a previous study by Tomas (2011, Cell Metabolism) showed a role for mitochondria-derived ROS. The role of ROS in mediating could easily be determined by treating cells with antioxidants. Beyond this, investigating the molecular pathway linking loss of Mfn1 to enhanced adipogenesis is beyond the scope of this paper, in my opinion.

Minor comments:

Adipogenic potential may decrease with passages. The experiment should have been set up so that differentiation was induced at the same passage.

It is surprising to see no expression of PPAR gamma or FABP4 at day 10 of differentiation in Mfn2-/- cells as these proteins are also expressed in undifferentiated cells.

Were there differences in proliferation?

6. PLOS authors have the option to publish the peer review history of their article (what does this mean?). If published, this will include your full peer review and any attached files.

Reviewer #1: No

Reviewer #2: No

---

## [Author Response · Author response to Decision Letter 0]

4 Mar 2024

Dear Dr. Monaco,

Thank-you for the opportunity to revise and resubmit this manuscript. We are also very grateful to the reviewers for their generally supportive comments and helpful suggestions. 

Our responses to each of the points made are shown below. All authors have seen the revised manuscript. We have added Ms. Pushpa Pushpa as a co-author, reflecting the additional work required as part of this Revision.

All original, uncropped blots are available in Supporting Information as 'Source data'.

We hope that the revised manuscript addresses your suggested revisions.

Yours sincerely

David

-----------

Reviewer #1: This manuscript has been carefully reviewed by 3 reviewers, whose comments are well aligned. The authors performed experiments to address some of the reviewers’ comments, but many key questions remained unanswered.

1, Characterization of mitochondrial functions in 3T3-L1 and/or MEFs as suggested by the reviewers would provide significant mechanistic insights. The analysis was presented in many publications. However, little has been done to address the differential regulation of mitochondrial functions by Mfn 1 and 2.

● Thank you for this suggestion. However, as the Reviewer indicates, work describing the differential regulation of mitochondrial function by Mfn1 & Mfn2 has been reported by others including the Chen group who originally generated the KO MEFs1–4. 

● Consequently, to make a significant contribution and advance the mechanism in the present study we would need to perform studies in differentiated adipocytes. We have attempted to quantify ROS production in the KO MEFs, but have not been able to generate reliable, reproducible and what we are persuaded are robust indices of ROS. This limitation is acknowledged in the revised submission.

● We have also elaborated our discussion to include potential future experiments.

● We have focused our efforts during this revision on the use of a ROS scavenger in modulating differentiation across the knock-out cell lines, thereby indirectly informing the differential regulation of mitochondrial ROS generation by the mitofusins in adipogenic differentiation.

2, Utilization of additional siRNA in 3T3-L1 cells and/or another line of Mfn1/2 KO MEF is recommended if re-expression of Mfn1/2 is not performed in KO MEFs as suggested by the reviewers.

● We have used three separate Invitrogen Silencer Select Mfn1 siRNA – one of these was associated with excess cell death whereas the other two resulted in similar phenotypes (see Rebuttal Letter Figure 1, below), though were most effective in combination (as reflected by the greatest knockdown).

● For Mfn2, in addition to use of a Dharmacon SmartPool Mfn2 siRNA, we also tested an Invitrogen silencer select siRNA for Mfn2 with similar observations (see Rebuttal Figure 2 below). As use of Dharmacon SmartPool Mfn2 siRNA led to the greatest reduction in target gene expression, we used data from this experimental paradigm in the manuscript.

● Unfortunately, no other Mfn1 or Mfn2 KO MEF lines are commercially available as far as we know.

Rebuttal letter Figure 1. Effect of two different anti-Mfn1 siRNAs (Invitrogen Silencer Select) on adipogenic differentiation of 3T3-L1s. A & B, Relative mRNA expression of markers of adipogenesis at day +10 differentiation (A). B, Lipid accumulation at day + 10 as measured by fluorescence using AdipoRed. D, Representative images for adipogenic differentiation using brightfield microscopy and AdipoRed staining. Data from n=2 biological repeats.

● The above example data shows the effect of two siRNAs against Mfn1 separately on 3T3-L1 differentiation. Since these different siRNAs had similar effects, we proceeded with “01 + 02” as a pool for Mfn1.

Rebuttal letter Figure 2. Effect of Silencer Select anti-Mfn1 and Silencer Select anti-Mfn2 siRNAs on adipogenic differentiation of wild-type MEFs.

● The above Oil Red O image is from using an Invitrogen silencer select for Mfn2, which is distinct from the siRNA to that reported in the paper, which was a Dharmacon SmartPool.

● Together, we conclude that the observations from knock-down studies are consistent across multiple different siRNAs, and generally consistent with the knock-out data.

Additional comments:

1, Mfn2-/- adipocytes have significant lower levels of GLUT1 and 4 than Mfn1-/- adipocytes (Fig 2D), but have higher basal rate of glucose uptake (Fig 2C). Please explain. Moreover, the insulin effect on glucose uptake is weak in the adipocytes, indicating that the differentiation procedure is not optimal.

● We also find this observation puzzling. We had also wondered whether the changes were due to Glut1 expression but did not observe consistent differences that would account for the finding.

● One hypothesis is that high basal glucose uptake in Mfn2-KO MEFs may be driven by dependence on glycolysis. These cells have a higher basal respiratory rate4. Differences in dominant source of ATP (glucose vs. lipid) in Mfn knock-out lines has been previously shown5,6.

● Mfn2 KO MEF do differentiate relatively poorly, certainly compared to 3T3-L1 cells. A smaller effect of insulin is further evidence of poorer differentiation of Mfn2 (also see PPARG expression western blot), but is challenging to interpret due to higher baseline. i.e. is the smaller fold change due to having higher basal.

● We have added further commentary on this observation in our discussion.

2, Quantification of lipid accumulation of Mfn1-/- adipocytes (day 4 vs day 8) in fig 2B is inconsistent with the images in fig 2A. Combination of the two panels in fig 2B allows the comparison of lipid accumulation during adipogenesis.

● Differentiation of WT MEFs is patchy (especially at day +8), which makes the comparison of WT and MFN2 KO lines visually challenging. AdipoRed provides an additional, less biased method of assessment. Whilst the data in 2B allows the comparison, we feel there is benefit from including the Oil Red O (2A) to give readers an overview of the differentiation phenotype.

3, Ln 151, concentrations of the stock solutions (sodium pyruvate, non-essential amino acids, penicillin/streptomycin) should be provided.

● Thank you. This has been done.

Reviewer #2: This study showed that loss of Mfn1 increases adipogenesis, while loss of Mfn2 decreases the expression of adipogenic markers without notable differences in lipid droplet. The authors suggest reactive oxygen species (ROS) as a potential mediator of the increased differentiation observed in Mfn1-/- cells. A role for ROS in stimulating adipogenesis was recently demonstrated by Singh et al. (Antioxidants and Redox Signaling, 2023) and a previous study by Tomas (2011, Cell Metabolism) showed a role for mitochondria-derived ROS. The role of ROS in mediating could easily be determined by treating cells with antioxidants. Beyond this, investigating the molecular pathway linking loss of Mfn1 to enhanced adipogenesis is beyond the scope of this paper, in my opinion.

● We have treated wild-type, Mfn1-/-, and Mfn2-/- MEFs with N-acetylcysteine (NAC) throughout differentiation and found that NAC inhibited adipogenesis in WT MEFs whereas it had minimal impact on Mfn1-/- MEFs. In our view, this could either be because ROS are not mediating the enhanced adipogenesis in Mfn1 null MEFs or because NAC used in the concentrations we used was not sufficient to prevent the impact of ROS in Mfn1 null cells. We have now included these data and discussed them in the revised manuscript. These data are now included as Figure 5.

Minor comments:

Adipogenic potential may decrease with passages. The experiment should have been set up so that differentiation was induced at the same passage.

● All cells were obtained at Passage zero from ATCC and experiments were performed with cells at passage <10. It is unclear precisely how many passages were performed by ATCC prior to us receiving the cells and therefore it is not possible to know in comparison to studies done by McFie et al.

● We have clarified the passage number in Methods.

It is surprising to see no expression of PPAR gamma or FABP4 at day 10 of differentiation in Mfn2-/- cells as these proteins are also expressed in undifferentiated cells.

● Mfn2-/- MEF did express lower levels of multiple adipogenic markers, even though they showed similar lipid accumulation to WT (fig 2B). Similar results were observed in 3T3-L1 KD. We have clarified in the discussion that this is indicative of poorer differentiation.

● In Figure 2D the wild-type cells are all differentiated adipocytes. We have made this more clear in the legend.

Rebuttal letter Figure 3. Immunoblot for markers of adipogenic differentiation in MEFs: wild-type (undifferentiated, grown to day +8), wild-type (differentiated), Mfn1-/- (differentiated), and Mfn2-/- (differentiated). Representative of n>5 replicates.

● In the above representative blot (Rebuttal letter Figure 3), we can observe that undifferentiated MEFs do not express PPAR gamma or Fabp4, or, at least, at a substantially reduced level compared to differentiated cells.

Were there differences in proliferation?

● Mfn2-/- MEFs grew very rapidly, whilst Mfn1-/-Mfn2-/- (‘double’) KO and Opa1-/- grew slowly – all as previously reported7. We optimised cell culture protocols to ensure same passage and confluence in differentiation experiments comparing different cell line.

References

1. Chen, Y., Liu, Y. & Dorn, G. W., 2nd. Mitochondrial fusion is essential for organelle function and cardiac homeostasis. Circ. Res. 109, 1327–1331 (2011).

2. Legros, F., Lombès, A., Frachon, P. & Rojo, M. Mitochondrial fusion in human cells is efficient, requires the inner membrane potential, and is mediated by mitofusins. Mol. Biol. Cell 13, 4343–4354 (2002).

3. Chen, H. et al. Mitofusins Mfn1 and Mfn2 coordinately regulate mitochondrial fusion and are essential for embryonic development. J. Cell Biol. 160, 189–200 (2003).

4. Kawalec, M. et al. Mitofusin 2 deficiency affects energy metabolism and mitochondrial biogenesis in MEF cells. PLoS One 10, 1–18 (2015).

5. Scheideler, M. & Herzig, S. Let’s burn whatever you have: mitofusin 2 metabolically re-wires brown adipose tissue. EMBO reports vol. 18 1039–1040 (2017).

6. Boutant, M. et al. Mfn2 is critical for brown adipose tissue thermogenic function. EMBO J. 41, e201694914 (2017).

7. Chen, K. H. et al. Role of mitofusin 2 (Mfn2) in controlling cellular proliferation. FASEB J. 28, 382–394 (2014).

---

## [Decision Letter · Decision Letter 1]

26 Apr 2024

PONE-D-23-15779R1Loss of Mfn1 but not Mfn2 enhances adipogenesisPLOS ONE

Dear Dr. Savage,

Thank you for submitting your manuscript to PLOS ONE. After careful consideration, we feel that it has merit but does not fully meet PLOS ONE’s publication criteria as it currently stands. Therefore, we invite you to submit a revised version of the manuscript that addresses the points raised during the review process. While one of the reviewers gave a positive opinion on your manuscript, the second reviewer still has minor concerns. Please address these concerns so your manuscript can finally be accepted.

We look forward to receiving your revised manuscript.

Kind regards,

Benedetta Ruzzenente

Academic Editor

PLOS ONE

Journal Requirements:

Reviewers' comments:

Reviewer's Responses to Questions

**Comments to the Author**

1. If the authors have adequately addressed your comments raised in a previous round of review and you feel that this manuscript is now acceptable for publication, you may indicate that here to bypass the “Comments to the Author” section, enter your conflict of interest statement in the “Confidential to Editor” section, and submit your "Accept" recommendation.

Reviewer #1: All comments have been addressed

Reviewer #2: All comments have been addressed

2. Is the manuscript technically sound, and do the data support the conclusions?

Reviewer #1: Partly

Reviewer #2: Yes

3. Has the statistical analysis been performed appropriately and rigorously? 

Reviewer #1: Yes

Reviewer #2: Yes

4. Have the authors made all data underlying the findings in their manuscript fully available?

Reviewer #1: Yes

Reviewer #2: Yes

5. Is the manuscript presented in an intelligible fashion and written in standard English?

Reviewer #1: Yes

Reviewer #2: Yes

6. Review Comments to the Author

Reviewer #1: The reviewer's comments have been appropriately addressed in general. However, the effects of the Mfn1 siRNA (01) and (02) are different without explanation (Rebuttal letter Figure 1). siRNA02 does not affect the expression of Plin1 and Adipoq and has minimal effect on Glut4 as compared to siRNA1. Is it due to different knockdown efficiency? The knock efficiency of the 2 siRNAs needs to be included and the rebuttal letter Figure 1 to be presented in the supplementary data.

Reviewer #2: (No Response)

7. PLOS authors have the option to publish the peer review history of their article (what does this mean?). If published, this will include your full peer review and any attached files.

Reviewer #1: No

Reviewer #2: No

---

## [Author Response · Author response to Decision Letter 1]

14 May 2024

Reviewer #1: The reviewer's comments have been appropriately addressed in general. However, the effects of the Mfn1 siRNA (01) and (02) are different without explanation (Rebuttal letter Figure 1). siRNA02 does not affect the expression of Plin1 and Adipoq and has minimal effect on Glut4 as compared to siRNA1. Is it due to different knockdown efficiency? The knock efficiency of the 2 siRNAs needs to be included and the rebuttal letter Figure 1 to be presented in the supplementary data.

Thank you for your suggestions. Yes, the differences observed are likely related to the relative knock-down efficacy. In optimisation, we tested three separate siRNAs, one showed very little efficacy and was not taken forwards. The other two (siRNA01 and siRNA02) demonstrated enhanced adipogenesis correlating with their relative ability to reduce expression of Mfn1.

These data have been included as Supplementary Figure 7, including a western blot showing the knock-down efficacy.

---

## [Decision Letter · Decision Letter 2]

15 Jun 2024

Loss of Mfn1 but not Mfn2 enhances adipogenesis

PONE-D-23-15779R2

Dear Dr. Savage,

We’re pleased to inform you that your manuscript has been judged scientifically suitable for publication and will be formally accepted for publication once it meets all outstanding technical requirements.

Kind regards,

Benedetta Ruzzenente

Academic Editor

PLOS ONE

Additional Editor Comments (optional):

Reviewers' comments:

Reviewer's Responses to Questions

**Comments to the Author**

1. If the authors have adequately addressed your comments raised in a previous round of review and you feel that this manuscript is now acceptable for publication, you may indicate that here to bypass the “Comments to the Author” section, enter your conflict of interest statement in the “Confidential to Editor” section, and submit your "Accept" recommendation.

Reviewer #1: All comments have been addressed

2. Is the manuscript technically sound, and do the data support the conclusions?

Reviewer #1: Yes

3. Has the statistical analysis been performed appropriately and rigorously? 

Reviewer #1: Yes

4. Have the authors made all data underlying the findings in their manuscript fully available?

Reviewer #1: Yes

5. Is the manuscript presented in an intelligible fashion and written in standard English?

Reviewer #1: Yes

6. Review Comments to the Author

Reviewer #1: The reviewer's comments have been appropriately addressed in general.

Minor comment:

Fig S7A showed knockdown of Mfn1 with very similar efficacy by all three siRNAs including the third anti-Mfn1 siRNA (s85003), while the main text (lines 218-220) stated that the third one had limited efficacy.

7. PLOS authors have the option to publish the peer review history of their article (what does this mean?). If published, this will include your full peer review and any attached files.

Reviewer #1: No

---

## [Editor Report · Acceptance letter]

22 Aug 2024

PONE-D-23-15779R2 

PLOS ONE

Dear Dr. Savage, 

I'm pleased to inform you that your manuscript has been deemed suitable for publication in PLOS ONE. Congratulations! Your manuscript is now being handed over to our production team.

Kind regards, 

on behalf of

Dr. Benedetta Ruzzenente 

Academic Editor

PLOS ONE